# Influence of Fire Exposition of Fibre-Cement Boards on Their Microstructure

**DOI:** 10.3390/ma16186153

**Published:** 2023-09-10

**Authors:** Krzysztof Schabowicz, Tomasz Gorzelańczyk, Łukasz Zawiślak, Filip Chyliński

**Affiliations:** 1Faculty of Civil Engineering, Wrocław University of Science and Technology, Wybrzeże Wyspiańskiego 27, 50-370 Wrocław, Poland; krzysztof.schabowicz@pwr.edu.pl (K.S.); tomasz.gorzelanczyk@pwr.edu.pl (T.G.); 2Instytut Techniki Budowlanej, Filtrowa 1, 00-611 Warszawa, Poland; f.chylinski@itb.pl

**Keywords:** fibre-cement boards, composites, exposure to fire, microstructural examination, scanning electron microscope, fibres

## Abstract

The diagnostics of materials, elements and structures after fire exposure are very complicated. Researchers carrying out such diagnostics encounter difficulties at the very beginning, e.g., how to map fire conditions. In this publication, the authors focused on the analysis of the fibre-cement composite used as facade cladding. The fibre-cement boards are construction products used in civil engineering. The fibre-cement boards are characterised by two phases: the matrix phase and the dispersed phase. The analysis of fibre-cement composite was performed using non-destructive methods. The use of non-destructive methods in the future will allow for the analysis of facades after fires without the need to obtain large elements, which will significantly reduce costs while increasing safety. The aim of the work was to determine internal changes in the microstructure of fibre-cement boards after exposure to fire. The degraded samples were compared with reference samples in the evaluation of the microstructure. An analysis was performed using a scanning electron microscope, images of backscattered electrons (BSE) and maps obtained using Energy Dispersive X-ray Spectroscopy (EDX), which allowed conclusions to be drawn. The observed changes were presented in the form of photos showing changes in the composition of the plates, and they were commented on. It should be noted that fire temperatures act destructively, and a number of changes can be observed in the microstructure. The results of the work indicate that, in the future, the use of non-destructive methods will make it possible to assess the degree of degradation of the façade after a fire.

## 1. Introduction

Fires present very dangerous situations, especially in the case of buildings used by people. They cause a lot of damage, which is also reflected in the elements, materials, structures, etc. After performing the rescue operation and extinguishing the fire, its diagnostics is important for the further survival of the building. The facades were usually completely dismantled after such an impact; the authors note that diagnostics using non-destructive methods will make it possible to determine whether these elements pose a threat. The authors focused on the analysis of fibre-cement boards using non-destructive methods which, in the future, may be used to diagnose real elements and facades.

The fibre-cement boards are construction products that have been used in civil engineering since the beginning of the 20th century. In the 1990s, the product was modified, and asbestos fibres that are dangerous to health were replaced with other fibres, mainly cellulose ones. Fibre-cement boards are composite materials consisting of two phases, including a continuous matrix phase surrounding the other phase, called a dispersed phase (strengthening elements). A schematic diagram showing the division of the phases is presented in Figure 1.

Fibre-cement boards are fibre-reinforced composite materials characterised by two phases. The first one is the matrix phase which, in the referenced board’s case, is a Portland-cement-based matrix responsible for matrix bonding and making it durable. The other one is a dispersed phase in the form of fibres. In addition to Portland cement, the matrix contains additional ingredients and fillers, i.e., lime flour, mica, pearlite, kaolin and microspheres [2]. The fibres in the dispersed phase are non-continuously distributed and randomly oriented. The fibres used in board manufacturing include cellulose fibres, PVA (polyvinyl alcohol) fibres and PP (polypropylene) fibres. Most manufacturers of fibre-cement boards use all the fibres mentioned above in the dispersed phase. Each fibre type fulfils a different role. Cellulose fibres form a spatial network reinforcing the whole composite, while PVA and PP fibres are added to improve the strength and durability of fibre-cement boards, especially those intended for outdoor use. Fibre-cement boards are intended for indoor and outdoor use. Outdoors, they are principally used for façade cladding and must meet the requirements set out by harmonised standards [3].

Research on fibre-cement boards in various environmental situations in the scientific literature is constantly being extended. In the field of controlling the current condition of the boards, there are many publications investigating these phenomena [4,5,6,7,8], but such methods cannot be applied to the analysis of burnt boards. The authors, having read the scientific literature, decided to use a non-destructive method for the diagnostics of fibre-cement boards—analysis using scanning electron microscope (SEM). The use of such a method results from the studies of other researchers who studied similar phenomena, such as Schabowicz [9], Verma [10] and Ngo [11].

High temperatures have a significantly degrading impact on most construction materials, including but not limited to fibre-reinforced composites. Scientific papers on fibre-cement boards exposed to fire focus on the evaluation of strength parameters [4]. 

The characteristics of fibre-reinforced composites in high-temperature conditions were investigated by references [12,13,14,15,16], who used stable heating as the source of fire—which does not match the testing method employed by this study’s authors—but many conclusions can be drawn from them.

Szymków [17] carried out tests and evaluated the possibility of re-using fibre-cement boards after exposure to high temperatures. Fibre-cement boards were not exposed to fire (flames) but were prepared as samples and heated in a furnace for some time. The samples were tested in two temperature groups, i.e., 230 °C and 400 °C, for various heating durations in the furnace. The samples heated in the furnace at 230 °C for three hours revealed a MOR (modulus of rupture or flexural strength) that was reduced by 70–80% versus the reference sample. Among the samples exposed to 400 °C for fifteen minutes in the furnace, only one out of five was suitable for performing three-point bending and determining the MOR. The other series lost their strength parameters in a shorter time at 400 °C. One should bear in mind that the temperature inside a furnace is more stable than when façades are exposed to flames in a non-controlled or controlled fire. Szymków [17] unequivocally shows the destructive impact of fire exposure on the load capacity of fibre-cement boards. The fibre-cement boards revealed no or minimum MOR and ED modulus of elasticity up to 1.6 m above the furnace (source of fire). The façade cladding elements from the area most exposed to high temperatures degraded and came off the substrate during the test; they did not survive the whole test, which lasted sixty minutes. Schabowicz et al. [8] analysed fibre-cement boards after exposure to fire in a real model, using a sound emission method during a three-point bending test. The samples revealed damage through high-energy brittle destruction at a low number of incidents (Nzd), which confirmed that the dispersed phase had been completely degraded. 

As described above, the dispersed phase may contain cellulose, PVA and PP fibres. The following melting points characterise each fibre type: PVA (polyvinyl alcohol) fibres—ca. 200–220 °C [18,19]; PP (polypropylene) fibres—ca. 175 °C [18,19,20]; and cellulose fibres—260–270 °C (decomposition) [21]. 

In the course of assessing the macrostructure of fibre-cement composites, the work of Gou [22] should be analysed. Gou [22] performed a similar method of obtaining samples and a similar type of analysis, but their analysis did not concern fibre-cement boards, but, more broadly, cellulose fibre-reinforced concrete composites (CFRC). The final conclusions were similar to other studies, whereby the boards burst without warning and the associated structures lost their loading capacity within a short time, but it is also worth analysing all the indirect phenomena that they observe. Concrete gradually degenerates with increasing temperature. At 200 °C, a small amount of visible water vapor escaped. With constant heating, calcium hydroxide—as the main cement hydrate in the concrete—gradually decomposes. At 600 °C, the concrete was almost dehydrated. The water evaporation and decomposition of the components cause mass loss in concrete. Concrete’s microstructure is damaged at high temperatures; water vapor pressure and thermal expansion inside the concrete will damage its microstructure. In addition, some cement hydrates (such as calcium silicate hydrate (C–S–H) gels) dehydrate and CH decomposes at a temperature of 800 °C. With heating at 1050 °C, rare bulk hydrates exist in the CFRC but are mainly transformed to glass; it should be noted that such temperatures do not occur on the façade.

The chemical reactions occurring in fibre-cement boards are another vital aspect related to the analysis of the boards’ chemical composition. At 105 °C, the free water in the matrix phase starts to evaporate quickly; in the temperature range between 80 and 150 °C, ettringite is dehydrated. The temperatures between 150 and 170 °C represent the gypsum dehydration range [21]. The temperature of ca. 300 °C makes chemically bound water evaporate, which reduces the material’s compressive strength. Portlandite decomposes between 400 °C and 540 °C. When the temperature increases over 400 °C, the cement matrix’s strength drops faster and decomposes due to the degradation of calcium silicate hydrates (C–S–H), forming β-C2S (β-dicalcium silicate) [21]. 

Velesick [23] presented that cellulose fibres burn at temperatures of up to 400 °C while the determined endothermic effects end at 389 °C with mass loss of 71–75%, depending on the type of cellulose used for armouring. This is very significant information, but in the case of the tests [23], the course of temperatures was constant, and not variable as in the case of the fire.

For fibre-cement boards, which represent composites, the temperatures degrading the boards’ structure (dispersed phase and matrix phase) and the impact of high temperatures on reducing the materials’ load capacity are known. Unfortunately, the scientific literature does not mention microstructure analysis in fibre-cement boards degraded by high temperatures. In the case of degradation covered by fire, the strength parameters can be evaluated using MOR tests [4], sound emission tests [24,25,26] or SEM structure analysis [17].

Having identified the deficiencies in the literature, the authors attempted to determine the impact of high temperatures on the microstructure change in fibre-cement boards and explain the causes of flexural strength changes using a scanning electron microscope (SEM).

## 2. Materials and Methods

Microstructure tests, on fibre-cement boards degraded due to exposure to high temperatures, were performed in the study. 

Samples were taken from the large scale facade model that was exposed to fire. The large scale facade model is a research platform to which the ventilated facade is attached. The research platform is used to study the impact and development of fire, which is a very good way to study the development of fire and to assess individual elements of the façade in terms of its flammability, the behavior of individual materials in fire conditions, or the fire safety of the entire façade system. Figure 2 shows the course of the test, including the impact of flames coming out of the hearth on the façade cladding.

Sample D3 was taken from the large-scale facade model that was exposed to fire. The sample was taken from an element that fell off at the 34th minute of the test. The locations of the item used for making the D3 sample for further tests are presented in Figure 3. The temperature course in D3 sources was recorded by thermocouples TE3 TE7; the results of thermocouples are presented in Figure 4. 

The degraded sample analysed in this paper was also subjected to tests using sound emission; the sample’s visual assessment was presented in papers by Ngo [11] and Szymków [17], demonstrating its structural destruction.

Polished cross-sections were immersed in resin as well as elements of D.ref (reference board, not exposed to fire) and D3 fibre-cement boards. The section samples are shown in Figure 5. Before the tests, the samples were vacuum-coated with gold using a Quorum model Q 150R ES coater. The vacuum coating of non-conductive materials with gold is necessary since scanning electron microscopy requires removing the electric charge from the analysed sample’s surface. Vacuum coating with gold is commonly used for analysing cement composite samples [27]. The samples were cut out from the reference board and from a façade cladding piece which came off the large scale façade model exposed to fire (shown in Figure 2 and Figure 3).

The examination was carried out using a Sigma 500 VP scanning electron microscope from Carl Zeiss (Jena, Germany), as shown in Figure 6. BSE images and Energy Dispersive X-ray Spectroscopy (EDX) maps were recorded during the examination. EDX works together with SEM to provide qualitative and semi-quantitative results. Both techniques together have the potential to introduce basic information about the material composition of scanned samples, which could not be obtained by other laboratory techniques.

For the sake of this paper, two microscope options were used, analysing the material’s microstructure and elemental composition. 

The elemental composition in the micro area and mapping were performed with an EDX detector, Ultim Max 40 from Oxford Instruments (Abingdon, UK), using AztekLive software v. 6.0 SP2. The device, including the detectors and the test chamber, is shown in Figure 7. 

The item from which the D3 sample was collected came off during the test in the ca. 34th minute. The maximum temperatures in this location ranged between 300 and 400 °C. 

It should be noted that the article may be an introduction to the diagnosis of façades that have been destroyed by fire.

## 3. Results

Figure 8 shows a summary of sample images of the D.ref and D3 samples’ microstructure. The microstructure observations were carried out on the polished cross-sections’ microstructure using a BSD (Backscattered Electron Detector), and mapping was performed with an EDX (Energy Dispersive Spectroscopy). The sections were observed with an SE (Secondary Electron) detector. 

Figure 9 summarises the share of elements in the reference and degraded samples’ structures. 

An analysis of the images shown in Figure 8 revealed the board’s compact structure, with visible cement binder grains on the D ref. sample’s sections. Air pores elongated in the direction parallel to the boards’ surface were present. The cellulose fibres present in the sample were typically parallel to the board’s surface. The structure’s apparent anisotropy results directly from the manufacturing process. The fibres’ arrangement in the structure favourably affects the composite’s flexural strength. In the D ref. sample, the board’s textured top layer was more compact than the substrate and relatively well-bound with the composite. Additionally, it was covered with a thin coat of paint on the outer side. The cellulose fibres observed on the fractures were well bound in the cement matrix and, as a result of the sample’s bending, most fibres were broken and pulled out of the matrix, which is a testimony to the fact that the fibres’ adhesion to the matrix fibres significantly exceeds the fibres’ tensile strength. 

During the tests on the real model, the D3 sample was exposed to temperatures between 300 and 400 °C for over thirty minutes. It revealed significant microstructure changes compared to the D ref. sample. The board’s textured and thin paint layers were completely degraded and delaminated. The degraded sample’s surface was not uniform and was characterised by protruding gaps. The tested composite’s porosity increased significantly, which could have been caused by releasing thermal decomposition gas products and the pyrolysis of some of the composite’s ingredients. At the temperature observed during the tests, one should expect a loss of non-bound and bound water from the C–S–H phase. Moreover, ettringite decomposition could occur, releasing significant amounts of steam. Furthermore, the portlandite contained in the structure could start to decompose, especially in the sample areas near the surface, where the temperature was higher. Cellulose fibres were subjected to pyrolysis throughout the board’s observed cross-section thickness, leaving empty voids inside the structure; the voids were sometimes filled with solid decomposition products. An analysis of the sections’ structures revealed no fibres that could transfer mechanical loads during bending. Only empty voids were observed in the cement matrix, left after the fibres; in a few cases, they were filled with decomposition products. 

Figure 9 shows the D.ref and D3 samples that were analysed using the BSD image. In terms of the share of the carbon element (C), its significant share is visible in the near-surface zone—in the hardener layer and in the paint coating. These layers were completely degraded by the high temperature in sample D3. In the case of the element oxygen (O), due to the degradation of the near-surface zone of the D3 sample, its smaller amounts are noticeable. The share of the element oxygen (O) in the degraded sample is much smaller due to its participation in the combustion process (chemical reactions), where, as a consequence, after its participation in the chemical reaction of combustion, the sample is weakened.

The above changes in the composition of the microstructure confirm that the fibre-cement board is subject to significant degradation and its usefulness is significantly limited.

## 4. Discussion

The impact of high temperatures on materials is intrinsically destructive. The time in which materials or products maintain the required properties is the high-temperature impact range. The maximum values of temperatures observed in the large scale façade model were from 500 °C to 700 °C. For the degraded sample analysed in the paper, the temperatures ranged from 300 °C to 400 °C. The destructive impact of high temperature on the tested fibrous cement boards’ structure becomes evident when analysing the results of a three-point bending test determining the flexural strength (MOR). Our own studies involved a composition and microstructure analysis of the reference samples and samples exposed to fire. The study was designed to explain the reasons for the observed sudden loss in the board’s elasticity in the temperature range from 300 to 400 °C.

The results of fibre-cement boards’ microstructural tests help explain the causes of the obtained flexural strength (MOR) values. Exposure to temperatures between 300 and 400 °C leads to irreversible microstructure changes, causing the relaxation and decomposition of cellulose fibres. Fibre decomposition directly contributes to the changes in the nature of the entire composite’s mechanical properties, including but not limited to the material’s response to bending force. A material deprived of fibres becomes more brittle as it contains no ingredient that could take some of the impacting force after a scratch is formed in the composite. This is also confirmed in other studies [22,23].

The first noticeable phenomenon at the temperature observed during the tests (300 to 400 °C) is the loss of non-bound and bound water from the C–S–H phase. In the case of furnace studies, this was also the first phenomenon to occur [22].

In addition, along with the duration of the fire, an increase in the carbon element (C), and a decrease in the oxygen element (O), can be observed.

The tests unequivocally revealed that fibre degradation leads directly to the changes in the nature of the whole composite’s mechanical properties, including but not limited to the material’s response to the bending force. A material deprived of fibres becomes more brittle, indicating high-energy fracture as the destruction method. Moreover, it shall be underlined that a microstructural analysis of fibre-cement boards enables determining the boards’ degradation rate and provides information about their re-use potential.

## 5. Conclusions

To conclude, the authors would like to emphasise that the tests in this study are significant from the point of view of building practice in general, as well as science, since previously, the literature made no mention of fibre-cement boards’ microstructure analysis and the changes after exposure to fire. An analysis of fibre-cement board’s microstructure enables the determining of the board’s degradation rate. The tests can be used independently or as part of a broader test scope in order to determine the board’s destruction rate. The most important conclusions are presented below:Exposure to temperatures between 300 and 400 °C leads to irreversible microstructure changes, causing the relaxation and decomposition of cellulose fibres. A material deprived of fibres becomes more brittle as it contains no ingredient that could take some of the impacting force after a scratch is formed in the composite.Fibre decomposition directly contributes to the changes in the nature of the entire composite’s mechanical properties, including but not limited to the material’s response to bending force.The first changes in the microstructure appear after the loss of the paint and texture coatings.Non-destructive SEM methods enable the diagnosis of fibre-cement boards after exposure to fire.

The conclusions presented above show that an analysis of the microstructure of fibre-cement boards confirms the degree of degradation of the boards, which can be used to diagnose the type after fires. 

Moreover, it should be highlighted that the presented tests are a part of broader scientific research. The authors also carry out tests on fibre-cement boards exposed to fire using other methods.

## Figures and Tables

**Figure 1 materials-16-06153-f001:**
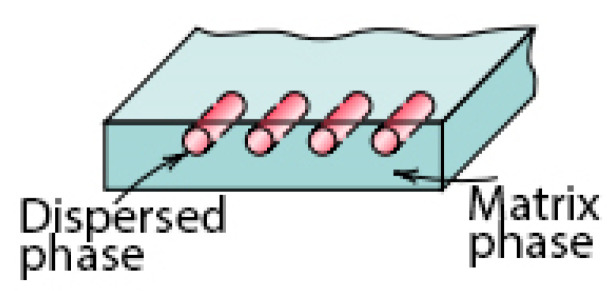
Two phases of composite materials [1].

**Figure 2 materials-16-06153-f002:**
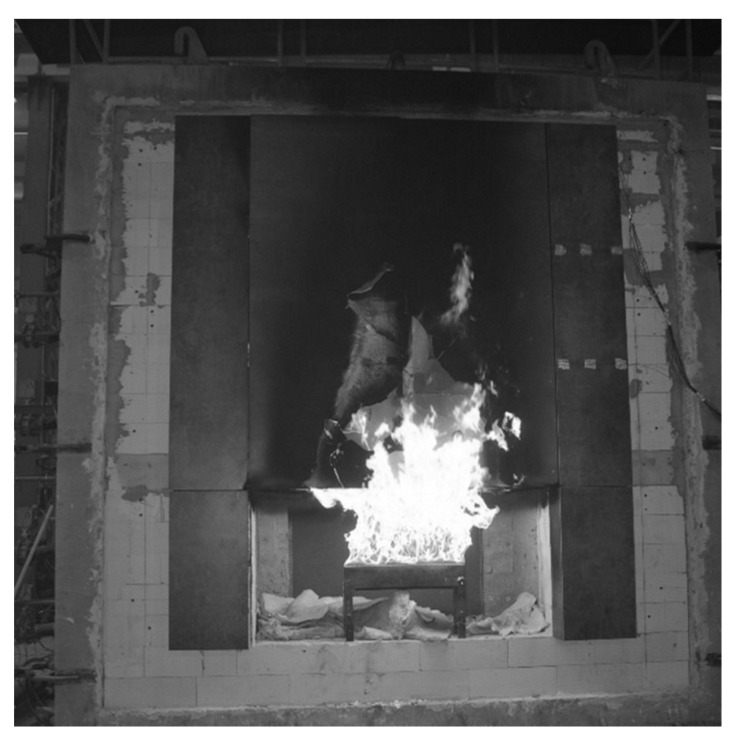
The large scale facade model during exposure to fire.

**Figure 3 materials-16-06153-f003:**
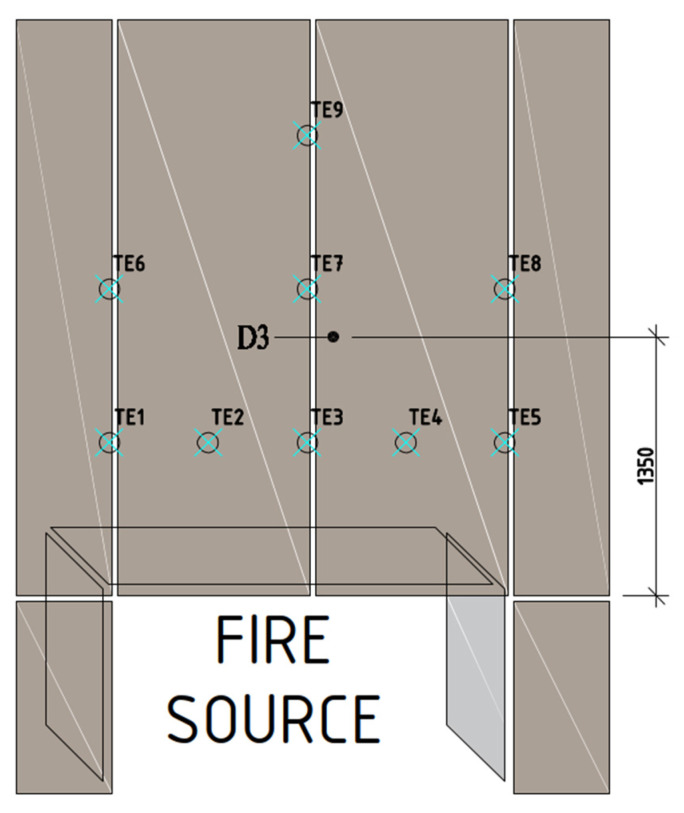
Diagram of external cladding board arrangement for fire impact analysis.

**Figure 4 materials-16-06153-f004:**
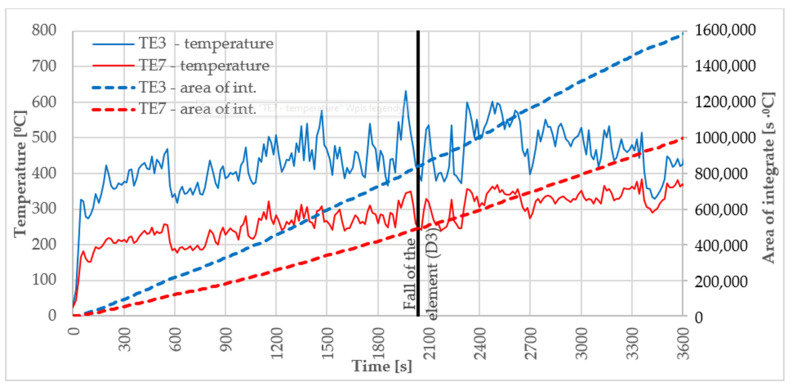
Temperature measurement results for TE3, TE7, thermocouples and the increasing integral for the temperature–time function, along with the determination of the fall-off time D3 element.

**Figure 5 materials-16-06153-f005:**
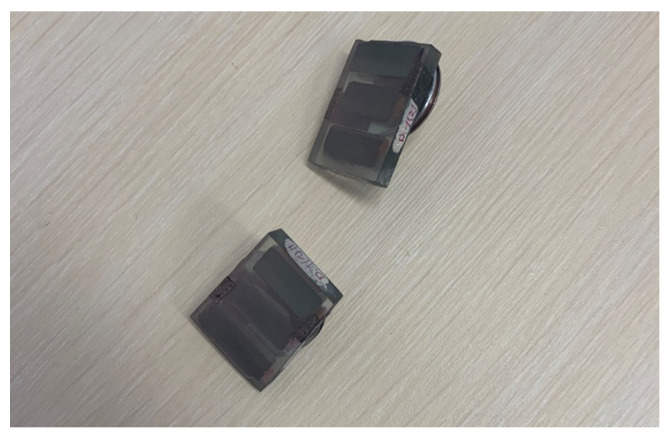
Section samples for scanning microscope examination.

**Figure 6 materials-16-06153-f006:**
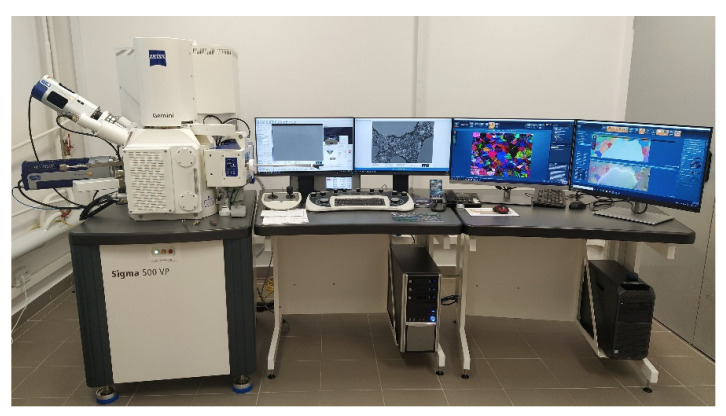
Scanning electron microscope from Carl Zeiss, Sigma 500 VP.

**Figure 7 materials-16-06153-f007:**
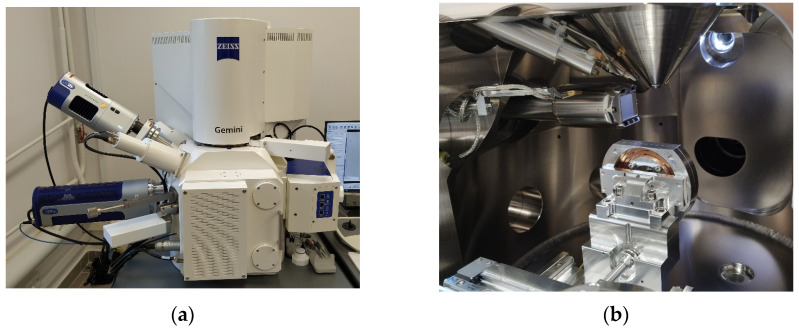
Sigma 500 VP scanning electron microscope from Carl Zeiss: (**a**) device, (**b**) test chamber.

**Figure 8 materials-16-06153-f008:**
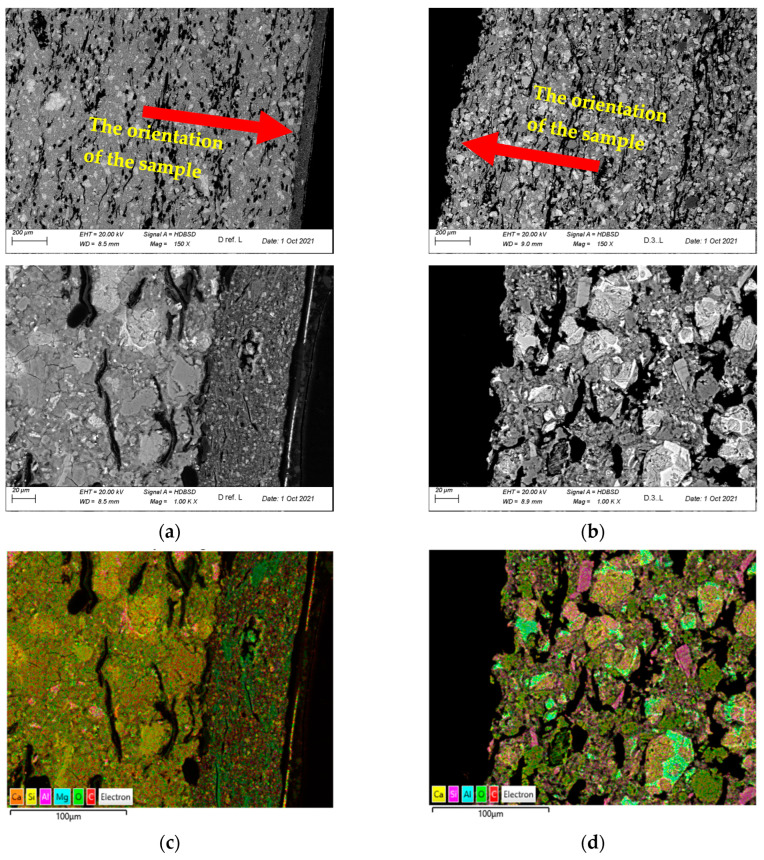
Summary of sample images of the analysed D.ref and D3 samples’ structures: (**a**) D.ref sample—section’s BSD images; (**b**) D3 sample—section’s BSD images; (**c**) D.ref sample—section’s EDX map; (**d**) D3 sample—section’s EDX map; (**e**) D.ref sample—fracture’s SE images; (**f**) D3 sample—fracture’s SE images.

**Figure 9 materials-16-06153-f009:**
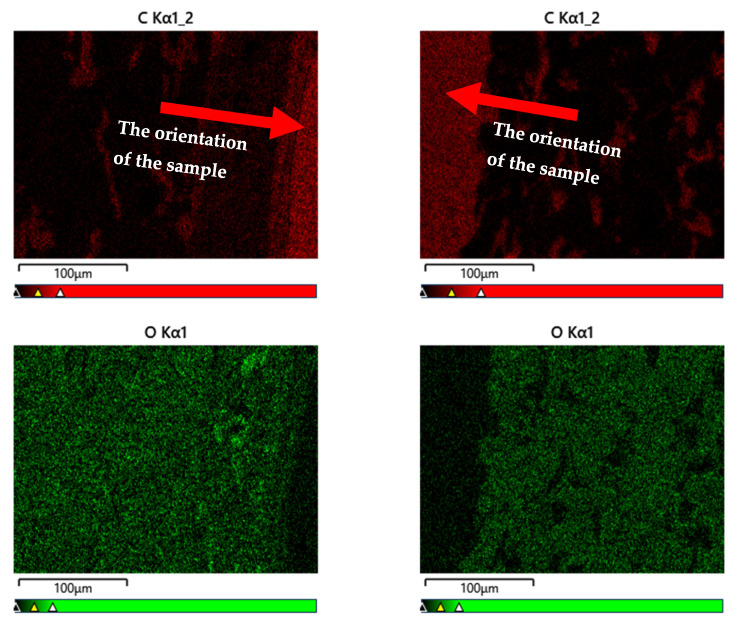
Summary of sample images of the analysed D.ref. and D3 samples’ structures: (**a**) D.ref sample—section’s BSD images; (**b**) D3 sample—section’s BSD images.

## Data Availability

Not applicable.

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
