# Peer review of "Influence of Fire Exposition of Fibre-Cement Boards on Their Microstructure"

_materials, 2023, doi:10.3390/ma16186153_

Round 1
Reviewer 1 Report (Previous Reviewer 1)
1- lines 20-21: “The aim of the work was to determine internal changes in the macrostructure of fiber cement boards after exposure to fire” but the title of the manuscript is “Influence of fire exposition of fibre-cement boards on their microstructure”. Which one is correct? “Macrostructure” or “microstructure”?
2- Add numerical results in the abstract and conclusion.
3- Why are there two conclusion sections (sections 4 and 5)? Merge sections 4,5 and 6 of the manuscript.
4-Rewrite conclusion point 4. Moreover, Rewrite lines 180-181.
5-Line 89: “Paper [10]” should be changed to “Szymków [10]”. Similar mistakes are seen in the manuscript.
6- Previous works have not been studied completely in the introduction section. Only, some references were added at the end of the sentences in the introduction. In fact, the introduction section has not been revised yet. The previous works should be added with details and findings, not only by adding references at the end of the sentences. For example, you can explain the works in papers number [16-20] with more details. There are many works about the fiber reinforced composites subjected to high temperatures that you can use. The introduction section needs deeply revision.
7- change references [2-4] with related published papers.
8- The results have not been discussed. They are only presented.
9- The references are not in a uniform format. For example: reference [27] is issue 2 but reference [28] is no.13. which one is correct? Issue or no.? please use the same format for all references
10-line 23: what is BSE?
1- The English is very weak. Consult a native speaker. It needs deeply revision. Some errors are as follows:
Lines 16 and 17: ” Fibre-cement” and “fiber”. Which one is correct? Fiber or fibre?
Line 16: “con-struction” should be changed to “construction”. Similar errors are seen in lines 268-271 and other parts of the manuscript.
Lines 16 and 17: “engineering: characterised by two phases: the..”. why two “:”?
Line 17: You use Britain English or American English in the manuscript? “characterised” or “characterized”? Which one is correct?
There are many errors like these examples.
Author Response
in attach

Reviewer 2 Report (Previous Reviewer 3)
Thanks for revising the article.
Author Response
We are deeply grateful to the Reviewer for the effort put in the review of our paper.
Round 2
Reviewer 1 Report (Previous Reviewer 1)
1- The references are not in a uniform format.
2- Rewrite conclusion point 4:” Internal microstructure changes….. high temperatures”.
3- Self citations are seen (references 2-4)
4- There are many works about the fiber reinforced composites subjected to high temperatures that you can use. The introduction section needs to be improved.
5-The results have not been discussed.
6- line 260: "Schabowicz [8]" but ref. [8] is "8. Verma, S. K"
Authors did not modify English. Errors are still seen such as line 268: “fi-bre”
Author Response
in attach

This manuscript is a resubmission of an earlier submission. The following is a list of the peer review reports and author responses from that submission.
Round 1
Reviewer 1 Report
1-Abstract has been poorly written.
2- Introduction is poor and previous works have not been studied completely. In addition, there are too many self-citations.
3- The procedures of tests have not been explained with details.
4- There is only summary section and there is no conclusion section in the manuscript.
5- There is no explanation about figure 8 in the manuscript.
6- The results have not been discussed.
Minor editing of English language required.
Author Response
Answer in attachment

Reviewer 2 Report
In this paper, authors investigated the impact of the fire exposition on flexural strength of fiber-cement composites. The article can be accepted for publication after the significant revisions according to the comments below:
1. The article is poorly organized. Please, structure and rename each chapter regarding to the journal rules.
2. The output of this article is based on SEM images and EDX maps. But authors indicated the mechanical property in the title. It would be better to correlate structure before and after degradation with mechanics, i.e. perform statistically relevant numbers of mechanical tests.
3. How many specimens did you study? Why did you analyze so small region of interest via EDX?
4. The summary of the article is obvious and very common. Of course, the structure will change and mechanics will decrease after fire.
Author Response
Answer in attachment

Reviewer 3 Report
The work is interesting but it looks to me that the background study was not investigated well. There are many studies in this area. This study has some new finding. But it is impossible to understand the motivation and improvement with comparison to the existing knowledge. Please discuss the literature more elaborately using more examples. Most, if not all, of the references used are local to authors. Please use some international references. One example, civil engineering materials - introduction and laboratory testing, by CRC Press. The conclusion section can be rewritten to concisely give the conclusion. In current state, this section became a discussion section. Or, a new conclusion section can be added.
Author Response
Answer in attachment

Round 2
Reviewer 1 Report
Unfortunately, the comments have not been responded correctly and in my opinion, the manuscript does not reach the required quality standard of the journal.
English should be improved throughout the manuscript